# An Actuator Surface Model to Simulate Vertical Axis Turbines

**Lucy Massie [1], Pablo Ouro [2], Thorsten Stoesser [1],* and Qianyu Luo [1]**

[1]  Department of Civil, Environmental and Geomatic Engineering, University College London,
    London WC1E 6BT, UK; lucy.massie.18@ucl.ac.uk (L.M.); qianyu.luo.18@ucl.ac.uk (Q.L.)
[2]  Hydro-environmental Research Centre, School of Engineering, Cardiff University, Cardiff CF24 3AA, UK;
    OuroP@cardiff.ac.uk
*   Correspondence: t.stoesser@ucl.ac.uk

**Abstract:** An actuator surface model (ASM) to be employed to simulate the effect of a vertical axis turbine on the hydrodynamics in its vicinity, particularly its wake is introduced. The advantage of the newly developed ASM is that it can represent the complex flow inside the vertical axis turbine's perimeter reasonably well, and hence, is able to predict, with a satisfying degree of accuracy, the turbine's near-wake, with a low computational cost. The ASM appears to overcome the inadequacy of actuator line models to account for the flow blockage of the rotor blades when they are on the up-stream side of the revolution, because the ASM uses a surface instead of a line to represent the blade. The ASM was used on a series of test cases to prove its validity, demonstrating that first order flow statistics—in our study, profiles of the stream-wise velocity—in the turbine's vicinity, can be produced with reasonable accuracy. The prediction of second order statistics, here in the form of the turbulent kinetic energy (TKE), exhibited dependence on the chosen grid; the finer the grid, the better the match between measured and computed TKE profiles.

**Keywords:** actuator model; large-eddy simulation; vertical axis turbines; LES; ASM; VAT

## 1. Introduction

A significant majority of climate scientists agree that it is extremely likely that changes to the global climate over the last century can be attributed to human activities [1]; particularly, increasing levels of greenhouse gas emissions to the atmosphere since the industrial revolution. A report by the IPCC (Intergovernmental Panel on Climate Change) in 2014 [2] states that "the energy supply sector is the largest contributor to global greenhouse gas emissions," highlighting the potential impact that advancements in sustainable, low/zero-emission energy production could have in the interest of mitigating humans' influence on the Earth's climate.

Tidal stream energy is a sustainable resource that benefits from being highly predictable; however, despite major technological breakthroughs in recent years, at present it is still a largely unexploited resource [3]. At the moment only one commercial tidal stream energy has been operational: SIMEC Atlantis Energy developed MeyGen in the Pentland Firth in Scotland, Europe's largest tidal power project and MeyGen's first phase (completed in 2017) included the design, manufacture and deployment of four 1.5 MW turbines. MeyGen employs horizontal axis turbines (HATs); HATs have received much of the attention in research and deployment in tidal energy projects thus far, probably due to the success of and experience gained from their wind energy counterparts. Recently, however, research focus has also been on vertical axis tidal turbines (VATTs) due to the advantages they offer when compared to HATs. Firstly, VATTs are omni-directional, meaning they operate with the same efficiency regardless of incoming flow direction; they also operate at lower tip-speed ratio,

thus, reducing noise and fish mortality [4]; and finally, the wake generated by a VATT recovers quickly, which is particularly beneficial in the context of turbine array design and optimisation [5].

One of the earliest experimental studies on VATTs, from Brochier et al. in 1986 [6], provided a concise description of vortex shedding of VAT blades in a dynamic stall through flow visualisation and velocity profile measurements. Subsequent contributions then built upon these findings to render a comprehensive picture of the processes involved in the formation and recovery of the wake of a rotating VAT [5,7–13]; to record quantitative data for use in numerical model validation [5,8,11,12]; and to investigate the performance of different turbines or turbine configurations [9,14]. The flow-dynamics have been shown to be strongly dependent on the tip-speed ratio (TSR) of the turbine, with the Reynolds number having little effect on the wake [7]. At a lower TSR, the turbine blades will undergo large angles of attack which results in greater flow separation on their inner sides leading to large vorticity shedding [11]; the blades will enter dynamic stall on the upstream side of their path, resulting in the shedding of two counter-rotating vortices, from the leading and trailing edges. These vortices are convected downstream by the mean flow, through the rotor's swept area and across the path of the blade, where the blade will interact with them, resulting in an increase in lift [6]. At a higher TSR, the larger relative velocity of the blades reduces the flow separation, and hence the role of the dynamic stall vortices on the turbine's hydrodynamics [4].

The kinematics of a turbine governs the extraction of energy from the mean flow, resulting in a velocity deficit behind the rotor, with a greater velocity deficit observed when it is operating at higher TSR [6,7]. The counter-rotating vortex-pairs behind a VAT play a key role in the re-energisation of the wake by transporting momentum from the unperturbed, higher velocity flow outside of the wake [10] into that wake. In addition, due to the complex dynamics of the vorticity field, a VAT's wake is substantially asymmetric, which increases with increasing distance from the rotor [12].

Computational fluid dynamics (CFD) is a valuable tool for the prediction of performance and hydrodynamics of VATs. High-fidelity models, such as a large-eddy simulation (LES) fluid solver coupled with the immersed boundary (IB) method, have been shown to accurately reproduce the flow physics, particularly in the wake, of a rotating VAT [4,15,16]. However, these types of models are computationally expensive, demanding a few hundred CPUs to deal with the associated fine meshes. Reynolds-averaged Navier–Stokes (RANS) simulations are less computationally expensive, but for the simulation of VATs, RANS models are not as suitable for capturing the flow separation experienced by the blades and dynamic stall vortices, even if the near-wall grid resolution is fine enough [17]. Studies which aim to simulate multiple turbines or which require a quick solution should make use of more simplified models that are still able to reproduce the hydrodynamics to a reasonable degree of accuracy.

The actuator line model (ALM) was introduced by Sørensen and Shen in 2002 [18], in which body forces acting on the rotor blades are determined and distributed along lines representing the blades, which are then projected back onto the flow field, enforcing the action of the blades on the fluid. The method was originally developed and validated for the simulation of HATs, but has since been successfully adapted to VATs [19–25]. The ALM benefits from being relatively computationally inexpensive; however, it has some short-comings and models often require multiple corrections; e.g., dynamic stall or end-effect corrections [19]. Mendoza and Goude [23] reported sensitivity to grid size and time step in their ALM simulations. Abkar and Dabiri [22] found in their validation case that the ALM was more capable of reproducing the hydrodynamics and velocity deficit of the far-wake, with some discrepancies in the near-wake. Bachant et al. [19] compared an ALM coupled with RANS simulations to that coupled with LES, and found that the LES was more accurate at reproducing the mean flow-field, but considerably under-predicted turbulent kinetic energy in the wake. Abkar [21] investigated the influence of the subgrid-scale (SGS) model on the results of an ALM and found that it is sensitive to choice of turbulence closure in LES.

Shen et al., later, in 2009 [26], extended their model to a 2D actuator surface model (ASM) for HATs, which they showed was able to accurately reproduce the resultant hydrodynamic forces for

their test case. The ASM has since been validated in other studies (e.g., [27]) and was shown to predict, with good accuracy, the near-wake velocity deficit and turbulent kinetic energy for a HAT, with the nacelle included as an actuator surface. To date, no record of an ASM having been implemented to simulate the hydrodynamic effect of an operational VAT, in a fully 3-dimensional flow field, can be found in the literature.

In the present study, an ASM, coupled with a LES fluid solver, is proposed and then employed with the goal to replicate the experiments of Bachant and Wosnik [5], and hence demonstrate its validity and accuracy in predicting the hydrodynamics and turbulence in the wake of a VAT in operation. The numerical framework is presented in Section 2 with details of the LES solver and current ASM methodology, the novelty of which lies in prescribing pre-determined coefficients of lift and drag which allow for the simplification of the model formulation, compared with previous ALMs. Section 3 outlines the experimental set-up of Bachant and Wosknik [5] and the computational set-up of the present study. The results are given in Section 4, where grid sensitivity is discussed, and experimental measurements of mean stream-wise velocity and turbulent kinetic energy in the near-wake are discussed, validating the proposed model, alongside an analysis of the flow field enacted by the ASM, particularly regarding the formation and recovery of the wake. Finally, Section 5 presents the conclusions drawn from this study and future work.

## 2. Numerical Framework

### 2.1. Large-Eddy Simulation

The fluid flow was resolved using Hydro3D, a well-validated LES research code [28–34], including geometry-resolved simulations of vertical axis turbines [4,15] and the validation of an ALM for HATs [35]. The governing equations resolved in Hydro3D are the spatially filtered Navier-Stokes equations for turbulent, incompressible, three-dimensional flow:

$$\nabla \cdot \mathbf{u} = 0, \tag{1}$$

$$\frac{\partial \mathbf{u}}{\partial t} + \mathbf{u} \cdot \nabla \mathbf{u} = -\frac{1}{\rho}\nabla p + \frac{1}{Re}\nabla^2 \mathbf{u} - \nabla \cdot \tau + \frac{\mathbf{f}}{\rho}, \tag{2}$$

where $\mathbf{u} = (u_x, u_y, u_z)^T$ and $p$ are the spatially filtered velocity and pressure; $\tau$ is the subgrid-scale (SGS) stress tensor, approximated by the Wall Adapting local eddy-viscosity (WALE) SGS model of [36]; $Re$ is the Reynolds number based on the turbine diameter, $D$, defined as $Re = UD/\nu$ with $\nu$ denoting the fluid viscosity; $\rho$ is the fluid density; and $\mathbf{f} = (f_x, f_y, f_z)^T$ is the force projected onto the flow field by the actuator surface model. The WALE SGS model is chosen due to its capability of predicting zero SGS-viscosity in areas of laminar flow. This is advantageous for moving boundaries and areas of zero velocity as is the case here, because no viscous near-wall damping is required.

The simulations are advanced in time using a standard fractional-step method, combining a fifth-order weighted essentially non-oscillatory (WENO) differences scheme to compute the convective fluxes and a second-order central differences scheme for the calculation of the viscous stresses. The correction of predicted velocities is achieved using the multi-grid technique to solve a Poisson pressure-correction equation. The formulation of the fractional step method, based on the two-step Runge-Kutta pressure-correction is as follows:

$$\frac{\tilde{\mathbf{u}} - \mathbf{u}^{l-1}}{\Delta t} = \frac{\alpha_l}{Re}\nabla^2 \mathbf{u}^{l-1} - \alpha_l \nabla p^{l-1} - \alpha_l \left[\mathbf{u}\left(\nabla \cdot \mathbf{u}\right)\right]^{l-1} \tag{3}$$

$$\tilde{\mathbf{u}}^* = \tilde{\mathbf{u}} + \mathbf{f}\Delta t \tag{4}$$

$$\nabla^2 \tilde{p} = \frac{\nabla \cdot \tilde{\mathbf{u}}^*}{\alpha_l \Delta t} \tag{5}$$

$$\mathbf{u}^t = \tilde{\mathbf{u}}^* - \alpha_l \Delta t \nabla \tilde{p} \tag{6}$$

$$p^t = p^{t-1} + \tilde{p} - \frac{\alpha_l \Delta t}{2Re} \nabla^2 \tilde{p}, \tag{7}$$

where $l = 1, 2$ gives the sub-step of the fractional step method, with $l = 1$ referring to values at the previous time-step, $t - 1$, and $\alpha_1 = \alpha_2 = 1/2$ are the Runge–Kutta coefficients. The predicted Eulerian velocities, before and after the force projection from the ASM, are $\tilde{\mathbf{u}}$ and $\tilde{\mathbf{u}}^*$ respectively, and $\mathbf{u}^{t-1}$ and $\mathbf{u}^t$ are the velocities at the previous and present time steps. The velocities are stored in a staggered fashion; i.e., a pressure field computed at the cell centres and velocities at the cell faces. The pressure and pseudo-pressure are denoted by $p$ and $\tilde{p}$, and $\mathbf{f}$ is the force projected onto the fluid by the ASM.

### 2.2. Actuator Surface Model Formulation

The actuator surface model (ASM) presented here is an extension of the ALM, where the blade is represented by multiple actuator lines each describing the vertical length of the blade, and the combination of lines describes the chord-line forming a surface. This representation of the blade aims to enable the actuator model to better represent the dynamic stall vortices that greatly dominate the flow dynamics in VATs. A 2-dimensional (2D) sketch of a three bladed VAT rotating counter-clockwise at a rotational velocity of $\Omega$, using the ASM blade representation, is shown in Figure 1.

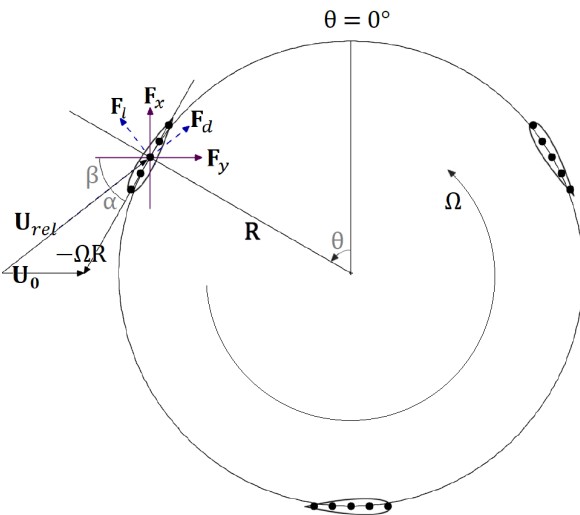

**Figure 1.** Actuator surface model (ASM) blade discretization, incident velocity vectors and configuration of the local ($F_d$, $F_l$) and global ($F_x$, $F_y$) force axes.

As the turbine rotates about its axis, lift and drag coefficients obtained from Ouro and Stoesser [4] are imposed to calculate lift ($F_l$) and drag ($F_d$) forces for each blade section as:

$$F_l = \frac{1}{2} \rho \, C_l \times Area \times (\Omega R)^2 \tag{8}$$

$$F_d = \frac{1}{2} \rho \, C_d \times Area \times (\Omega R)^2, \tag{9}$$

where *Area* is equal to $c \times \Delta z$, where $c$ is the chord length and $\Delta z$ is the mesh size in the vertical direction, $\Omega R$ is the rotational speed and $C_l$ and $C_d$ are coefficients of lift and drag, respectively, which are functions of the angle rotated by the blade, $\theta$, and account for the viscous and pressure contributions.

The lift and drag coefficients implemented in this study have the advantage of already accounting for the effect of dynamic stall and blade-vortex interactions seen in VATs, as they are obtained from geometry resolved simulations of a rotating VAT using LES-IB [4]. Figure 2 shows the lift and

drag coefficients determined in [4] along with the piece-wise polynomial functions employed in the calculations of the present study (see Appendix A.1 for details). For $330° \leq \theta < 360°$, the lift coefficient is capped at $C_l = -0.6$ to avoid excessive forcing being introduced during the last period of the rotation. These coefficients were readily available to the authors, and their suitability as estimations for the lift and drag coefficients for this case can be deemed reasonable since the wake dynamics downstream of VATs comprising similar NACA profiles are expected to change very little.

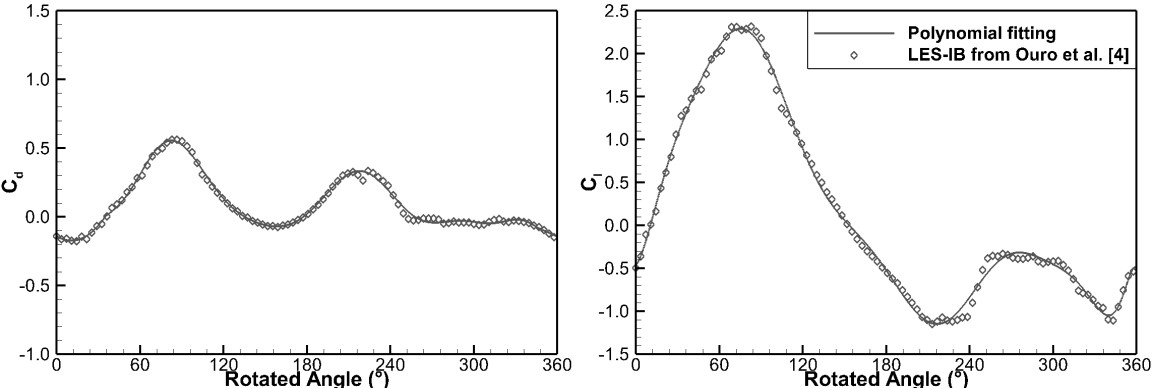

**Figure 2.** Coefficients of drag(**left**) and lift (**right**) plotted against rotated angle, showing the polynomial fit used in this study (solid lines) with coefficients (symbols) calculated from Ouro and Stoesser [4].

The hydrodynamic forces are initially calculated at the mass centre of the blade chord and are then distributed evenly across the actuator points describing the chord-line, to be projected back onto the flow field. To achieve this, $F_l$ and $F_d$ are translated from the rotational coordinate system of the turbine to the Cartesian coordinate system of the flow field, rotating the forces clockwise by the effective angle of attack, $\beta$, as shown in Figure 1. To determine this, first, the local velocity, $U^*$, at the actuator point at the blade centre of mass, is interpolated from the surrounding fluid cells via delta function $\phi_4$ Ref. [27]. The velocity vector is then translated to the turbine coordinate system, by rotating the vectors anti-clockwise by the rotated angle, $\theta$, to obtain the relative velocity, $\mathbf{U}_{rel}$, as:

$$\mathbf{U}_{rel} = \begin{pmatrix} U_t \\ U_p \end{pmatrix} = \begin{pmatrix} \cos\theta & \sin\theta \\ -\sin\theta & \cos\theta \end{pmatrix} \begin{pmatrix} U_x^* \\ U_y^* \end{pmatrix}, \tag{10}$$

where $U_x^*$ and $U_y^*$ are the interpolated local stream-wise and span-wise velocity components, and $U_t$ and $U_p$ are the velocity components tangential and perpendicular to the blade. From this, the local angle of attack is determined, according to [37], as:

$$\alpha = \arctan\left(\frac{U_p}{U_t + \Omega R}\right); \tag{11}$$

then, the effective angle of attack is calculated as $\beta = \theta - \alpha$. The blade forces can then be rotated to the Cartesian system as:

$$\begin{pmatrix} F_x \\ F_y \end{pmatrix} = \begin{pmatrix} \cos\beta & -\sin\beta \\ \sin\beta & \cos\beta \end{pmatrix} \begin{pmatrix} F_d \\ F_l \end{pmatrix}. \tag{12}$$

These forces are then projected back onto the surrounding fluid nodes via a spherical Gaussian function:

$$\eta = \frac{1}{\epsilon^3 \pi^{\frac{3}{2}}} \exp\left[-\left(\frac{|r|}{\epsilon}\right)^2\right], \tag{13}$$

where $|r|$ is the absolute distance between the actuator point and fluid node, and $\epsilon$ is the kernel width of the function, set to $\mathtt{MAX}\left(c/4, 4\sqrt[3]{V_{cell}}\right)$ as recommended in Mendoza et al. [24].

### 3. Test Case

The test case selected to demonstrate and validate the proposed ASM is that of the experiments presented in Bachant and Wosnik [5]. They performed tow tank experiments of a three-bladed VAT of height $H = 1$ m and diameter $D = 1$ m, similar to the Sandia/DOE Reference Model 2 (RM2), constructed from NACA0020 section blades with constant chord-length, $c = 0.14$ m. A constant towing speed of 1 m/s was set with the turbine rotating at tip speed ratio of $\lambda = D\Omega/2U_0 = 1.9$; i.e., $\Omega = 3.8$ rad/s. The turbine blades were mounted at half-chord and half-span, with struts of the same NACA profile, to a 0.095 m diameter shaft. Velocity measurements were taken by an acoustic Doppler velocimeter (ADV), one turbine diameter downstream of the turbine shaft for 270 different probe locations, providing a thorough data-set of the three velocity components in the near-wake.

*Computational Set-Up*

The dimensions of the computational domain replicate that of the tow tank in which the experiments were performed, with the stream-wise length reduced in order to reduce computational expense. Additional tests were conducted with an extended computational domain (approximately 50% longer in the $x$ direction) and it was found that there was no effect on the results. The lengths in the $x$, $y$ and $z$ directions were $L_x = 14.4\ D$, $L_y = 3.6\ D$ and $L_z = 2.4\ D$; details of the different grid resolutions investigated are given in Table 1, together with the number of actuator points per chord length. The centre of the turbine was placed 2.8 $D$ downstream of the inlet, in the centre of the $y - z$ plane. A uniform inflow velocity of $U_0 = 1$ m/s was imposed at the inlet, and a convective boundary condition was employed at the outlet. At the lateral walls and bottom of the domain, a constant wall speed of 1 m/s was prescribed to mimic the relative movement of the turbine in the towing tank, and at the top of the domain a slip condition was used to approximate a shear-free water surface.

**Table 1.** Spatial and temporal discretization of the simulations, and number of actuator points per chord length.

| Grid | $\Delta x, \Delta y, \Delta z$ | $\Delta t$ | *Points/c* |
|---|---|---|---|
| Coarse | $0.060D$ | 0.00700 s | 2 |
| Medium | $0.040D$ | 0.00325 s | 3 |
| Fine | $0.025D$ | 0.00125 s | 5 |

The turbine blades are represented using the ASM described in the previous section, with the rest of the rotor's structure omitted for simplicity. The turbine rotates in the anti-clockwise direction and simulations were run for a total of 20 revolutions, with first-order statistics being averaged after four revolutions and second-order statistics being averaged after eight revolutions. Simulations were run on a workstation that hosts 24 Intel Xeon X5620, 2.67 GHz cores. Details of the computational cost of the simulations are given in Table A1 in Appendix A.2.

### 4. Results and Discussion

The results, in terms of the rotor's wake, are provided at $x - y$ planes at three different vertical elevations of the domain: at the turbine mid-height, $z/H = 0.0$; away from the centre-plane close to the top of the blades, $z/H = 0.38$; and slightly above the turbine, $z/H = 0.63$. These elevations were selected from the available experimental data obtained from [5] in order to demonstrate a three-dimensional picture of the flow field generated by the ASM. First, a comparison of simulations employing different grid resolutions is presented in the discussion of Figures 3–5; all subsequent results presented are those obtained using the finest grid resolution; i.e., $\Delta x, \Delta y, \Delta z = 0.025\ D$.

Figure 3 shows a comparison of mean velocity profiles for three different grid sizes at elevations $z/H = 0.0$, 0.38 and 0.63, demonstrating the sensitivity of the ASM to grid resolution. Good agreement with experimental data was achieved on all three grids at elevations $z/H = 0.0$ and 0.38, with some discrepancies in the velocity gradients at the periphery of the wake on the $y/D < 0$ side. The local

velocity peak measured in the centre of the wake for $z/H = 0.38$ was not replicated, particularly on the coarse ($\Delta x = 0.06 \, D$) and medium ($\Delta x = 0.04 \, D$) grids; on the finest grid there was suggestion of this local peak but it was not captured. Above the turbine's tips at $z/H = 0.63$, the width of the wake was larger and the local velocity peak in the centre of the wake was less well-defined with increasing grid size. Differences in the results calculated on the different sized grids are most apparent at this elevation; although the overall predictions of velocity deficit are still in fairly good agreement with the experimental values. Some of the grid dependency exhibited here can be related to the backward force projection via the spherical Gaussian function, particularly due to the choice of the width parameter, $\epsilon$. For example, on the coarse grid $\epsilon = 4\sqrt[3]{V_{cell}} = 0.24 \, D$; therefore, the area over which the forces are projected is large and the effect of the forcing on the velocities is diffused. Further investigation into the optimal kernel width for force projection for the ASM would be beneficial to the future development of the model.

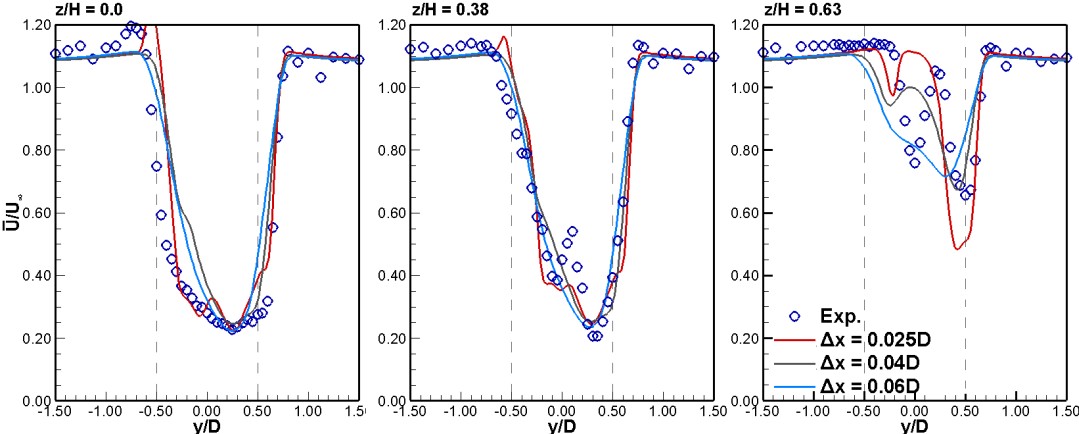

**Figure 3.** Mean stream-wise velocity profiles of the simulations(solid lines) together with experimental data (open circles) from [5] at $z/H = 0.0$ (**left**), $z/H = 0.38$ (**middle**) and $z/H = 0.63$ (**right**), for three grid resolutions.

Figure 4 presents simulated profiles of the LES-predicted turbulent kinetic energy (TKE) together with experimental data. Here, TKE was calculated from the sum of the square of velocity fluctuations, given by $TKE = 1/2(\overline{u'}^2 + \overline{v'}^2 + \overline{w'}^2)$. For all grids and at all elevations, the level of kinetic turbulence was greatly under-predicted. Clearly, the LES on the coarse grid predicts hardly any turbulence; however, the TKE peak at $y/D \approx 0.5$ was captured by the finest resolution. This indicates that the proposed ASM with prescribed lift and drag coefficients is able to represent, in some form, the dynamic stall vortices during the down-stroke motion of the blades. The grids employed here are relatively large for LES; thus, most of the turbulence is not resolved, and as will be discussed, the ASM is sensitive to choice of SGS model.

Figure 5 presents contours of the ratio of sub-grid scale (SGS) viscosity to fluid kinematic viscosity, at the mid-plane of the turbine for the three different grid resolutions employed. Values presented here are, overall, fairly large; this is due to the large filter widths (equivalent to the grid size) employed in the simulations and the presence of large local instantaneous velocity gradients. This is clear in the vicinity of the turbine, where dynamic stall vortices are present (locally very high drag and lift forces), and further afield downstream in the far-wake, where there are large gradients of velocity due to larger flow structures. On the coarse grid, there are a few large pockets of high SGS viscosity from approximately 6–7 $D$ downstream of the turbine, preceded by a region of low SGS viscosity in the wake. As the grid resolution increases, the intensity of the high $\nu_{SGS}/\nu$ pockets reduce due to smaller structures being resolved, until, on the fine grid, the patches are notably smaller. This indicates that

resolutions of at least $\Delta x/D = 0.04$ are required to enable the ASM to adequately represent small scale flow features.

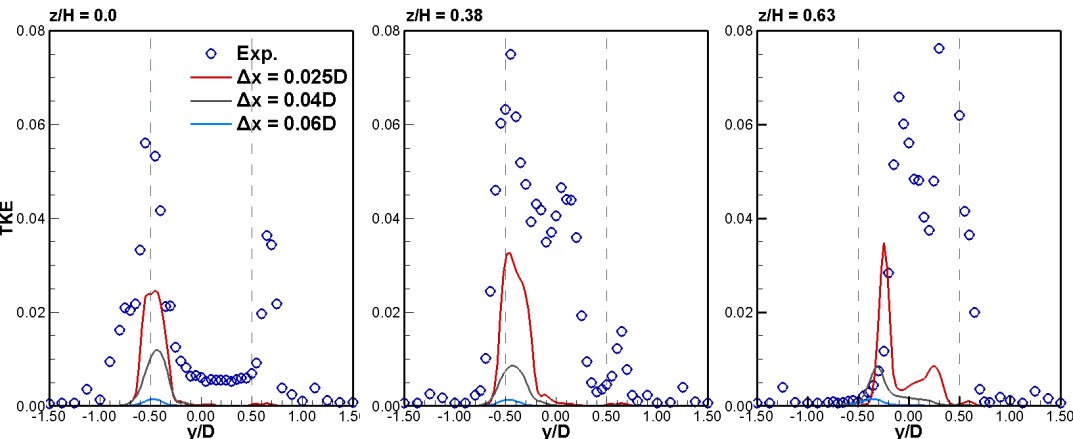

**Figure 4.** Profiles of turbulent kinetic energy (TKE) from simulations(solid lines) together with experimental data (scatter points) from [5] at $z/H = 0.0$ (**left**), $z/H = 0.38$ (**middle**) and $z/H = 0.63$ (**right**), for three grid resolutions.

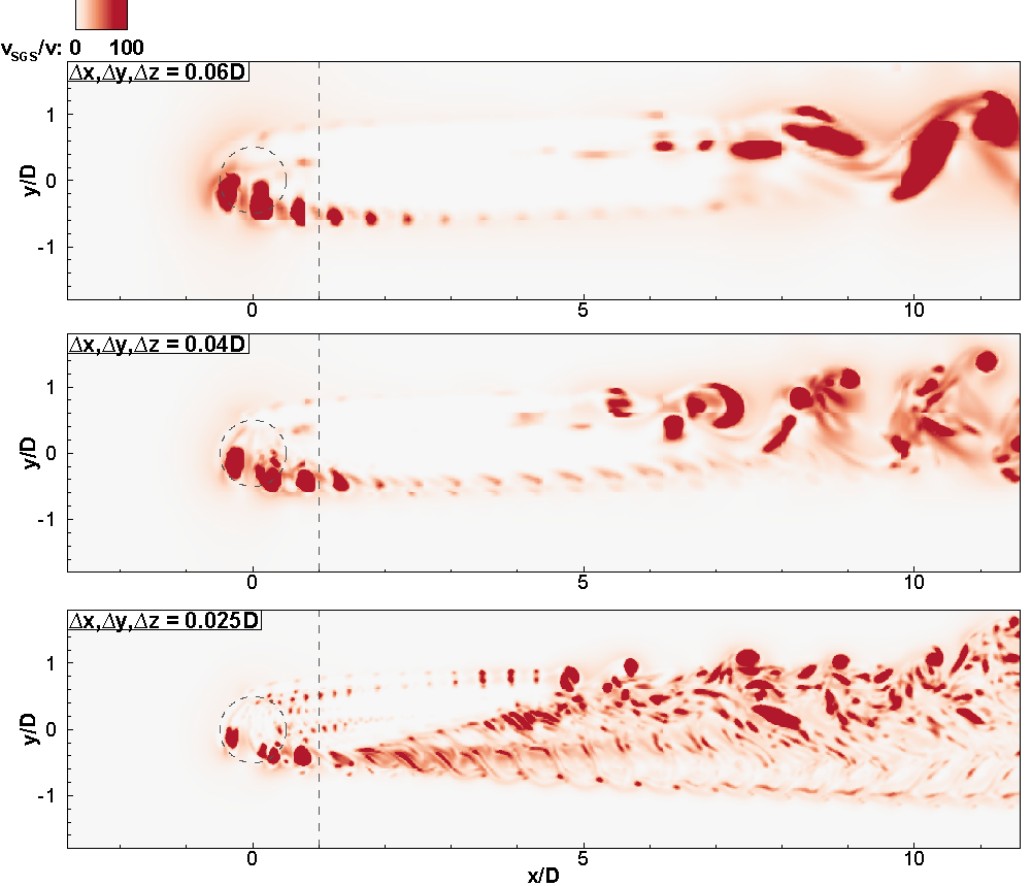

**Figure 5.** Contours of the ratio of sub-grid scale viscosity to fluid kinematic viscosity ($\nu_{SGS}/\nu$) for three grid resolutions, coarse (**top**), medium (**middle**) and fine (**bottom**).

Figures 3–5 may suggest that further grid refinement could improve the predictions of the model, and a fourth simulation was indeed carried out on an even finer grid than the "fine grid", and a slight

deterioration of the predictions of the first order statistics, together with a slight improvement of the second order statistics, was observed. The tendency of the simulations to not converge to a single solution is, however, not a surprise: As the grid is refined, the number of actuator points describing the surface along the chord-line increases and more details are resolved, but more detailed input is required. In fact, this "finest grid" would already allow blade-resolved simulations; however, the approach employed here is not designed for such highly-resolved simulations, as the detailed distribution of lift and drag forces along the chord is non-uniform and is in reality not known. Most importantly, the proposed model was designed with the goal of obtaining approximate predictions of the wake behind a VAT whilst keeping computational expense low; thus, employing a grid much finer than those presented in this study defeats the purpose.

The velocity contour at the centre-plane ($z/H = 0.0$) in Figure 6 shows that the near-wake features a considerable skewness towards the $y/D > 0$ direction, downstream of the up-stroke blade motion. Looking further downstream, the asymmetric and three-dimensional nature of the wake is demonstrated as it expands laterally and velocities recover more quickly on the $y/D < 0$ side. At $z/H = 0.38$ and above the turbine at $z/H = 0.63$, this skewness becomes increasingly less prominent, particularly at distances of 3–4 diameters downstream.

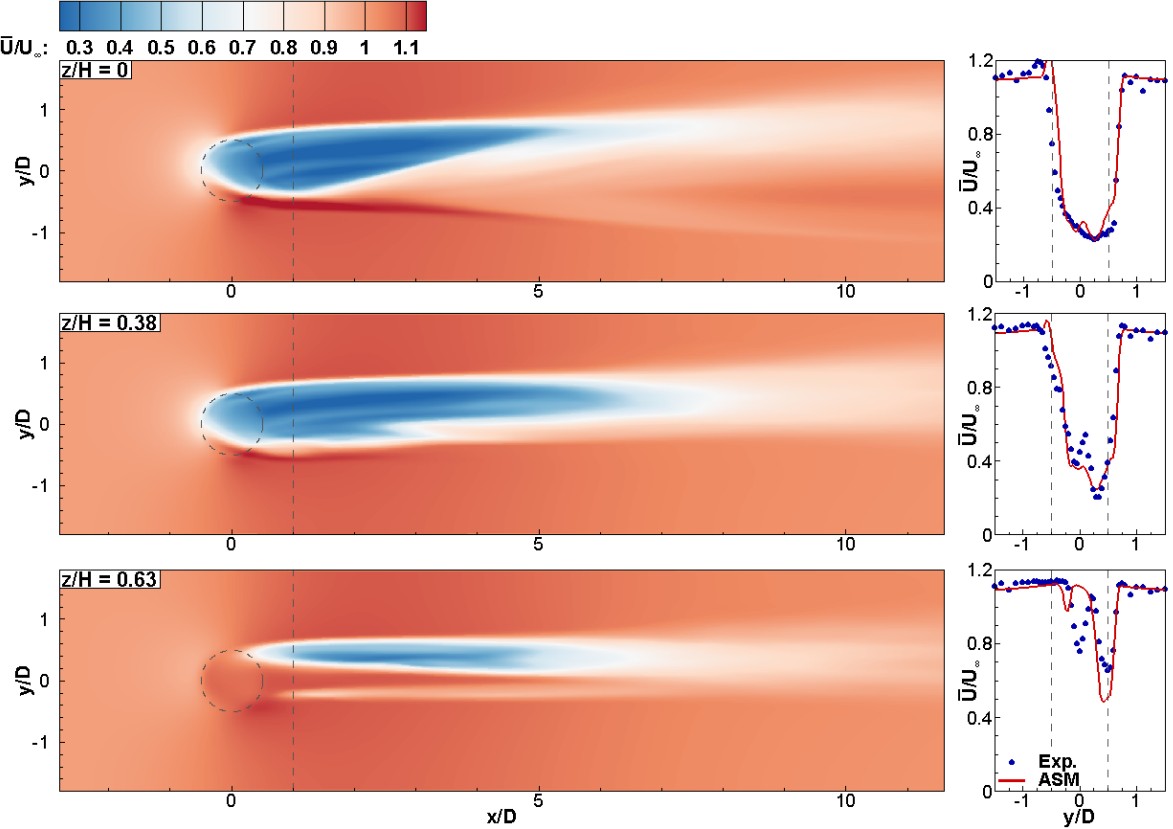

**Figure 6.** Left: Mean stream-wise velocity contours. Right: mean stream-wise velocity profiles, computed on the fine grid, together with experimental data from [5] at $x/D = 1$. Elevations $z/H = 0.0$ (**top**), $z/H = 0.38$ (**middle**) and $z/H = 0.63$ (**bottom**).

The mean stream-wise velocity profiles, presented on the right side of Figure 6 show close agreement with the experimental measurements of [5]. The velocity peak present at the edge of the wake at $z/H = 0.0$ is over-predicted in magnitude and shifted towards the centre of the domain, causing the wake to be slightly more narrow than that measured in the experiments; the situation is similar for $z/H = 0.38$. This is likely the result of excess force being introduced by the ASM around $\theta = 180°$; the cause of this is probably in the simplified calculation of forces. Here, the projected

area is constant at $c \times \Delta z$; however, in reality the frontal area changes with rotated angle. At $\theta = 0°$ and $180°$, this is $\Delta z$ multiplied by the width of the blade; i.e., 20% of $c \times \Delta z$ for a NACA0020 airfoil. The cap introduced on $C_l$ for $330° \leq \theta < 360°$ reduces the influence of this on the velocities on the $y/D > 0$ side but can be seen for $y/D < 0$. An additional effect of this is that on the down-stroke, the extra forcing causes the flow through the turbine to deflect excessively towards $y/D > 0$, resulting in the predicted wake being more narrow, as seen in the velocity profiles, and the skewness seen in the contours.

The contours of TKE, presented in Figure 7, show high levels of TKE produced in the turbine's vicinity, at $z/H = 0.0$ and $0.38$, coinciding with the shedding of dynamic stall vortices on the upstream side and the blade–vortex interaction on the downstream side of the turbine (see Figure 8). Between $x/D = 1$ and $5$, TKE decreases across the vertical extension of the wake. Thereafter, there is another pocket of high TKE on the $y/D > 0$ side of the wake, generated due to the stream-wise velocity fluctuations.

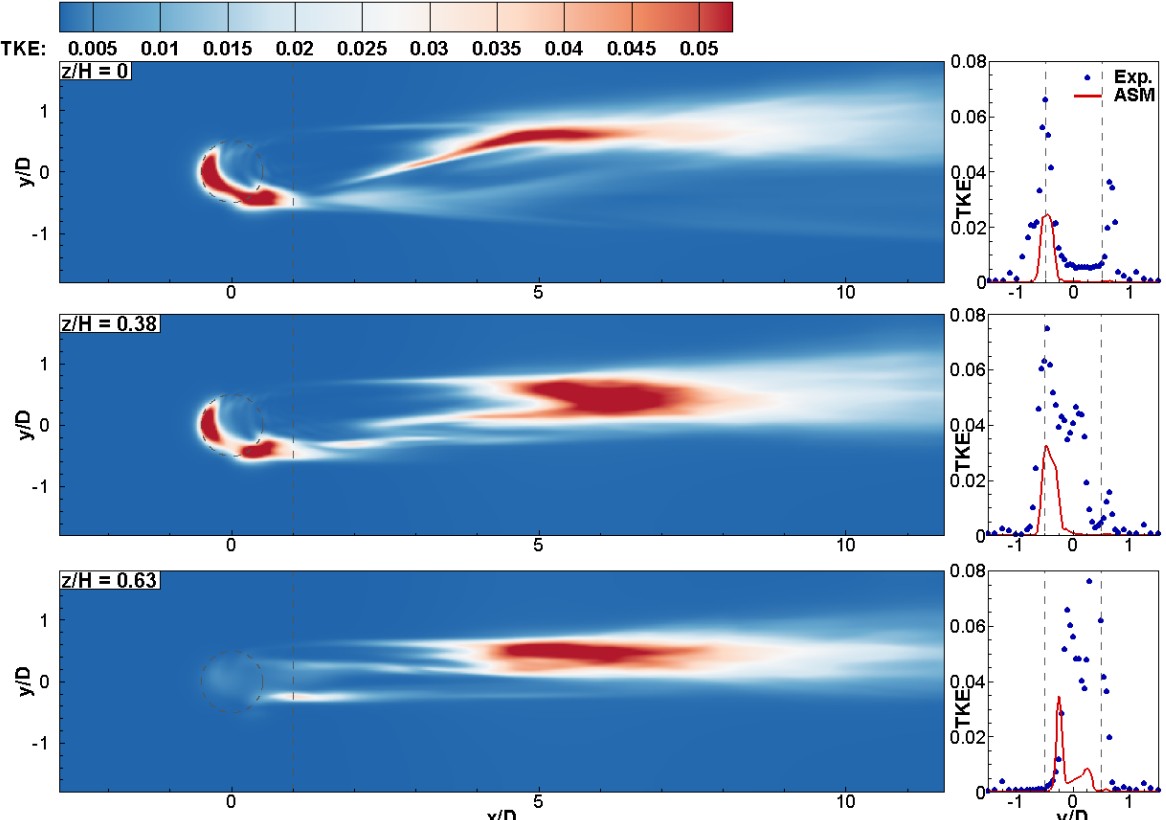

**Figure 7.** Left: Turbulent kinetic energy (TKE) contours; right: TKE profiles together with experimental data from [5] at $x/D$=1. Elevations: $z/H = 0.0$ (**top**), $z/H = 0.38$ (**middle**) and $z/H = 0.63$ (**bottom**).

The profiles on the right side of Figure 7 generally show that the ASM under-predicts turbulence, which can likely be attributed to the limited accuracy when representing flow phenomena, such as flow separation on the blades and dynamic stall, which can be captured using LES-IBM [4]. The locations of the peaks on the $y/D < 0$ side are in good agreement with the experimental data; the turbine-induced turbulence here is captured since it originates from the shedding of vortices. However, when $y/D > 0$, there is almost no turbulence produced; this is because the vortices shed by the ASM on this side of the turbine remain stable until they pass further downstream, where they eventually transition into turbulence, as demonstrated in the contours of Figure 8. Abkar [21] investigated the effect of SGS modelling and found that the results of ALM-LES were sensitive to the turbulence closure employed,

suggesting that this is a common issue in actuator-type methods and the selection of SGS model should be carefully considered.

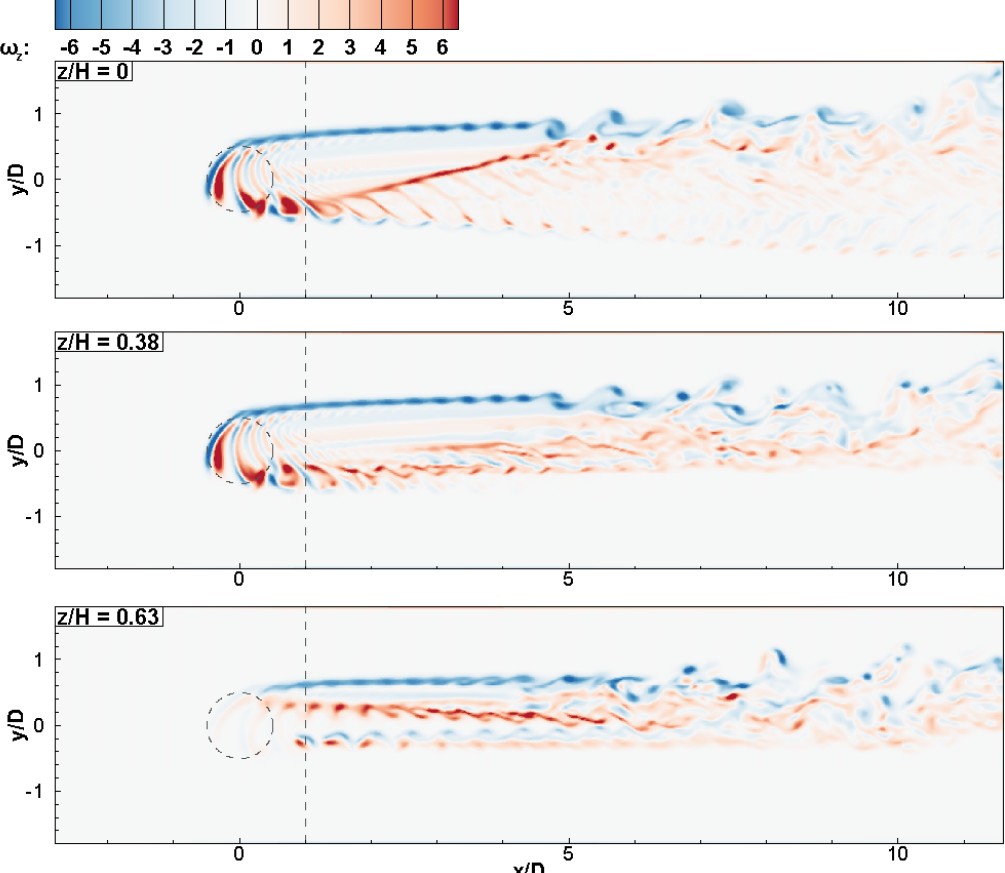

**Figure 8.** Contours of the vertical vorticity at elevations $z/H = 0.0$ (**top**), $z/H = 0.38$ (**middle**) and $z/H = 0.63$ (**bottom**).

Contours of the vertical vorticity, $\omega_z$, are presented in Figure 8. Blue and red contours, i.e., areas of high positive and negative vorticity, represent shed dynamic stall vortices and their interaction with the blades in the vicinity of the rotor. The motion of the turbine blades on the down-stroke generate the dynamic stall vortices, subsequently deflecting the wake towards the $y/D > 0$ side, as previously mentioned, and demonstrated most clearly at $z/H = 0.0$. The vorticity induced by the vortex-blade interaction deflects in the opposite direction and diffuses laterally as the wake recovers. At the elevations $z/H = 0.0$ and $0.38$, on the $y/D > 0$ side, a region of negative vorticity trailing from the location where the blade's rotated angle, $\theta = 0°$, corresponds to that of the missing peak in TKE values in Figure 7. These results correlate well with the fact that this wake region appears to be absent of vertical structures, which indicates the ASM is unable to represent the flow separation during the up-stroke motion of the blades. Above the turbine, at $z/H = 0.63$, the three-dimensionality of the wake is again demonstrated as the vorticity generated by the turbine expands in the vertical direction in the near-wake, increasing into the far-wake. Regularity of the vorticity is seen in the near-wake region, due to the periodic nature of the shedding and absence of ambient turbulence, then becoming increasingly irregular as it moves into the far-wake and transitions into turbulence at a distance of approximately $4D$, corresponding to regions of high TKE seen in Figure 7. In this region, the resultant increase in mixing recovers momentum from the unperturbed flow outside of the wake.

Figure 9 shows mean stream-wise velocity profiles at $z/H = 0.0$ and $0.38$, at different downstream locations comparing the proposed LES-ASM with the experimental data from Ouro et al. [13], who investigated the three-dimensional dynamics of the wake behind a three-bladed Gorlov VAT.

The experiments of [13] were carried out under turbulent flow, with the turbine operating at a similar tip speed ratio but higher flow blockage than the simulated case; thus, this experimental case is a different turbine operating in a (slightly) different flow. Hence, the comparison is to be seen more qualitative than quantitative. The rate at which the wake momentum recovers in both the ASM and the experiments presented here are in quite good agreement, except at $x/D = 5$ where the ASM overestimates the velocity deficit. This is probably due to the absence of ambient turbulence in the simulations which would aide in momentum mixing, and hence, quicker wake recovery. Overall, these results suggest that the ASM is able to reproduce the processes governing wake recovery, observed in experiments and in other CFD models. Here, the profiles at $x/D = 10$ show the closest agreement which indicates that the far-wake is replicated with more success than the near-wake, similarly to the findings of Abkar and Dabiri [22]. Since the intended application of the ASM is the assessment of turbine array design, a good representation of both the far-wake and wake recovery is a particularly positive outcome.

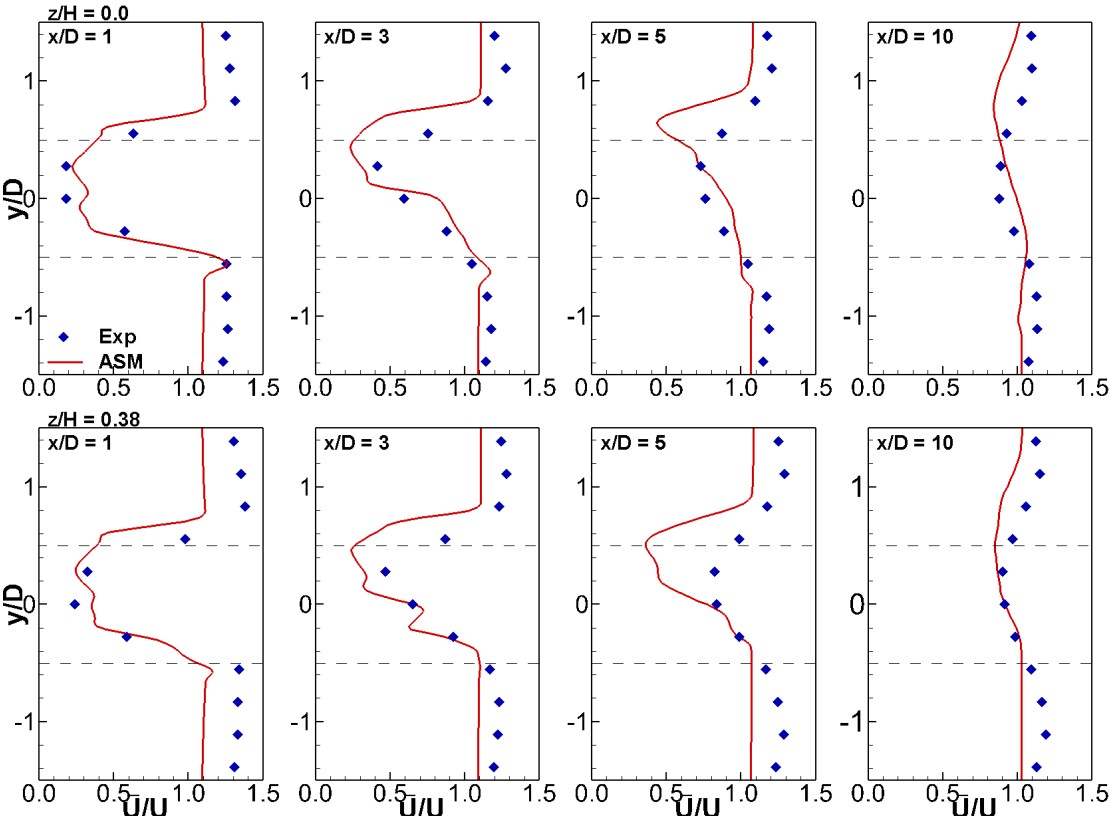

**Figure 9.** Mean stream-wise velocity profiles at downstream locations from left to right: $x/D = 1$, $x/D = 3$, $x/D = 5$ and $x/D = 10$, at elevations $z/H = 0.0$ (**top**) and $z/H = 0.38$ (**bottom**), compared with experimental results from [13].

## 5. Conclusions

An actuator surface model (ASM) was implemented in the in-house large-eddy simulation code Hydro3D and validated with experimental data. The proposed ASM uses prescribed lift and drag coefficients obtained from geometry-resolved large-eddy simulations to implicitly account for flow separation and dynamic stall effects on the blades. The comparison of predicted hydrodynamics in the form of time-averaged stream-wise velocity and turbulence in the form of the turbulent kinetic energy in the rotor's wake with experimental data, demonstrate that the ASM is capable of capturing the velocity deficit in the near wake with relatively good accuracy. However, the predicted levels of turbulent kinetic energy (TKE) in the wake are underestimated, in particular, when using coarse grids.

This is owing to the fact that the ASM (or, as has been demonstrated in past research, actuator line models) is unable to resolve local, large-scale flow structures that occur at the blade scale. Whilst it is shown that grid refinement improves the predictions of first and second order statistics, this trend cannot and will not continue when finer and finer grids are employed because ASM (and ALM) models have been designed to operate at low computational costs (and coarser grids), and hence should (and can) not replace blade-resolved simulations. The ASM presented here is able to capture the asymmetric nature of a VAT's wake in the horizontal plane as well as the vertical wake expansion, despite the fact that the ASM does not resolve tip vortices (which are, again, only resolvable in computationally expensive blade resolved simulations). The major advantage of the ASM is its low computational cost together with its reasonable accuracy in forecasting turbine wakes, which will allow future applications to wind and tidal turbine farm developments and planning at full scale. This will be explored in the future.

**Author Contributions:** L.M. implemented the ASM, ran the simulations and wrote the original draft of the manuscript. P.O. developed the concept of the ASM and reviewed the manuscript. Q.L. provided experimental data for and assisted with Figure 7. T.S. supervised the work of L.M. and reviewed the manuscript.

**Funding:** This research received no external funding.

**Acknowledgments:** The authors acknowledge the support of Supercomputing Wales, a project partly funded by the European Regional Development Fund (ERDF) via the Welsh government.

**Conflicts of Interest:** The authors declare no conflict of interest.

**Appendix A**

*Appendix A.1*

High-order polynomial functions were employed to approximate the lift and drag coefficients in this study. The lift coefficient, $C_l$, reads:

$$
\begin{aligned}
C_l(\theta) = & -9.83109403445010 \times 10^{-30} \times \theta^{14} + 2.35506917977834 \times 10^{-26} \times \theta^{13} \\
& -2.49085514318290 \times 10^{-23} \times \theta^{12} + 1.52861741834597 \times 10^{-20} \times \theta^{11} \\
& -6.01348602214706 \times 10^{-18} \times \theta^{10} + 1.57992317569433 \times 10^{-15} \times \theta^{9} \\
& -2.79676685658241 \times 10^{-13} \times \theta^{8} + 3.27393735916174 \times 10^{-11} \times \theta^{7} \\
& -2.39204712729446 \times 10^{-9} \times \theta^{6} + 9.28650148346635 \times 10^{-8} \times \theta^{5} \\
& -5.91984268776261 \times 10^{-7} \times \theta^{4} - 9.47384394282790 \times 10^{-5} \times \theta^{3} \\
& +3.14028704853696 \times 10^{-3} \times \theta^{2} + 2.19265827600333 \times 10^{-2} \times \theta - 0.470319202693645,
\end{aligned}
$$

and the drag coefficient, $C_d$, is calculated by:

$$
C_d\left(\theta\right) =
\begin{cases}
\begin{aligned}
& 2.96862876671061 \times 10^{-4} \times \theta^2 - 6.39641994261401 \times 10^{-3} \times \theta \\
& -0.140115557534005,
\end{aligned}
& \text{if } 0° \leq \theta < 35°, \\[2em]
\begin{aligned}
& -7.81195089028066 \times 10^{-34} \times \theta^{16} + 2.28728899458862 \times 10^{-30} \times \theta^{15} \\
& -3.03482604943372 \times 10^{-27} \times \theta^{14} + 2.41321554737758 \times 10^{-24} \times \theta^{13} \\
& -1.28157461368188 \times 10^{-21} \times \theta^{12} + 4.79472187285026 \times 10^{-19} \times \theta^{11} \\
& -2.29899815564960 \times 10^{-16} \times \theta^{10} + 2.57937811840519 \times 10^{-14} \times \theta^{9} \\
& -3.75662265252506 \times 10^{-12} \times \theta^{8} + 3.97565875154755 \times 10^{-10} \times \theta^{7} \\
& -2.99730852027196 \times 10^{-8} \times \theta^{6} + 1.55841955992838 \times 10^{-6} \times \theta^{5} \\
& -5.31865863851434 \times 10^{-5} \times \theta^{4} + 1.10154286382894 \times 10^{-3} \times \theta^{3} \\
& -1.18896450651628 \times 10^{-2} \times \theta^{2} + 5.01729242562662 \times 10^{-2} \times \theta \\
& -0.194577074144036,
\end{aligned}
& \text{if } 35° \leq \theta < 350°, \\[2em]
-4.87756197364469 \times 10^{-3} \times \theta + 1.61032763683459 & \text{if } 350° \leq \theta < 360°.
\end{cases}
$$

*Appendix A.2*

The details of the computational costs of each simulation presented in this study are given in the following table. Simulations were run on a workstation hosting 24 Intel Xeon X5620 @2.67GHz cores.

**Table A1.** Computational costs of each simulation presented in this study.

| $\Delta x, \Delta y, \Delta z$ | $\Delta t$ | Time-Steps to Complete 20 Revs | Wall-Clock Time Per Time-Step | Total Wall-Clock Time |
|---|---|---|---|---|
| 0.060 $D$ | 0.00700 s | 4800 | 0.2 s/$\Delta$t | 1039.27 s |
| 0.040 $D$ | 0.00325 s | 9500 | 0.6 s/$\Delta$t | 6995.15 s |
| 0.025 $D$ | 0.00125 s | 26,500 | 2.2 s/$\Delta$t | 59,602.99 s |

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
