# Peer review of "An Actuator Surface Model to Simulate Vertical Axis Turbines"

_energies, doi:10.3390/en12244741_

Round 1

Reviewer 1 Report

This article presents an ASM to simulate the effect of a wind turbine. The article is well written, however I cannot see what is the main contribution of the authors. Major comments:

1) Implementing a model that has already been implemented by others in a in-house code is not something for publication. Could you explain what is the novelty of your research? In case you do not introduce something new, you could extend your article explaining something related to the flow physics. You could study velocity fluctuations, describe the main flow structures, ...

2) The dimension of your domain in Lx seems too small (Lx=14D), but your comparison with the experimental results in the near field are sufficiently good. Could you comment something about how this model behaves in the far field? Why did you choose Lx=14D?

Reviewer 3 Report

The manuscript deals with introduction of a new method to accurately predict the complex flow inside vertical turbine. The authors report that the method is inexpensive.

The language of the manuscript should be significantly improved.

Does the ASM only consider the averaged lift and drag forces over the blades?

How does the model treat the the pressure drag and viscous drag over the blade?

Please plot your filter width size.

Please plot nut of SGS.

How did the authors calculate TKE.

Please give the details of the cost of the model.

I can not recommend the manuscript for publication in the Journal of energies. It needs major improvement

Round 2

Reviewer 1 Report

The authors have followed my suggestions. I recommend this article for publication.

Author Response

We thank the reviewer for their efforts in reviewing our revised manuscript. 

Author Response

We thank the reviewer for their efforts. 

Reviewer 3 Report

Hi,

The authors replied my comments.

I have two major points.

The flow with chord length of 0.14 m and velocity of 1 m/s is laminar. Right? firstly, how do you use WALE for such a flow? If my previous argument is right, how do you extrapolate your conclusion for other flow conditions in rotor-stator interaction? You say that the simulation with 8 million cells took 16 hours on 24 processors!! Is it from scratch or from the converged results? Please give a rough comparison with a full LES simulation from literature.

Author Response

We thank Reviewer #3 for their second review and we would like to respond to their comments as follows:

The flow with chord length of 0.14 m and velocity of 1 m/s is laminar. Right? firstly, how do you use WALE for such a flow? If my previous argument is right, how do you extrapolate your conclusion for other flow conditions in rotor-stator interaction?”

Indeed the developing boundary layer over the blade is laminar, however in our model we do not resolve the boundary layer but use "actuator forces" to pretend that there is a blade. Our method is developed to predict the wake of a VATT and not the blade-fluid interaction. The wake is fully turbulent, thus the need to employ a turbulence closure model for our LES. We use WALE for all flow conditions as it is a well-established sgs model that has proven to give valid sgs stresses in complex turbulent flows. The model is hence and (theoretically) valid to predict VATT flow interaction and its resulting wake.

“You say that the simulation with 8 million cells took 16 hours on 24 processors!! Is it from scratch or from the converged results? Please give a rough comparison with a full LES simulation from literature.”

It is from scratch, so basically half a day on a decent workstation. Our previous LES-IBM simulations (ref. 4 in the manuscript) required 171 CPUs to simulate a turbine of finite vertical length of only two chords. For a similar resolution, our case would require about 1,000 more CPUs running for 6 days, hence the cost would be 42x more CPUs for 9x more running time therefore requiring approximately 400x more computing effort.